# Glucose Addition Enhanced the Advanced Treatment of Coking Wastewater

**Na Li, Yu Xia \*, Xuwen He, Weijia Li, Lianhua Yuan, Xu Wu, Yihe Qin, Run Yuan and Xun Gong**

School of Chemical and Environmental Engineering, China University of Mining and Technology, Beijing 100083, China; m18811173132_2@163.com (N.L.); hexuwen@sina.com.cn (X.H.); 15266683348@163.com (W.L.); yuan_lh2020@163.com (L.Y.); xuwu202020@163.com (X.W.); qin775734416@163.com (Y.Q.); bqt2000302017@student.cumtb.edu.cn (R.Y.); gongxun6s@163.com (X.G.)
\* Correspondence: xiayu@cumtb.edu.cn; Tel.: +86-13683060988

**Abstract:** Biological processes have high removal efficiencies and low operational costs, but the secondary effluent of coking wastewater (CWW), even at a low concentration, is difficult for microorganisms to degrade directly. In this study, glucose was used as a carbon source and co-metabolic substrate for microbial acclimation in order to enhance the advanced treatment of coking wastewater (CWW). The removal performance of the pollutants, especially recalcitrant compounds, was studied and the changes in the microbial community structure after activated sludge acclimation were analyzed. The effect of glucose addition on the secondary biochemical effluent of coking wastewater (SBECW) treatment by the acclimated sludge was further studied by a comparison between the performance of two parallel reactors seeded with the acclimated sludge. Our results showed that the concentrations of chemical oxygen demand (COD), total organic carbon (TOC), and UV absorption at 254 nm ($UV_{254}$) of the wastewater decreased in the acclimation process. Refractory organic matter, such as polycyclic aromatic hydrocarbons and nitrogen-containing heterocyclics, in the SBECW was effectively degraded by the acclimated sludge. High-throughput sequencing revealed that microbes with a strong ability to degrade recalcitrant compounds were enriched after acclimation, such as *Thauera* (8.91%), *Pseudomonas* (3.35%), and *Blastocatella* (10.76%). Seeded with the acclimated sludge, the reactor with the glucose addition showed higher COD removal efficiencies than the control system without glucose addition ($p < 0.05$). Collectively, glucose addition enhanced the advanced treatment of coking wastewater (CWW).

**Keywords:** microbial acclimation; coking wastewater; advanced treatment; recalcitrant compounds; microbial diversity



## 1. Introduction

Coking wastewater (CWW) is a typical high concentration, organic wastewater generated in the coal refining, gas purification, chemical product recovery, and refining processes, that contains a variety of recalcitrant compounds [1,2]. Several pollutants remain in the secondary effluent, including organic compounds such as phenols, polycyclic aromatic hydrocarbons, and oxygen- and sulfur-containing heterocyclics [3,4]. Therefore, effective methods should be adopted to perform an advanced treatment on recalcitrant compounds after the secondary biochemical process for CWW.

At present, several methods have been developed for the advanced treatment of CWW, such as biological processes [5,6], adsorption [7,8], advanced oxidation [4,9,10], and membrane separation [11]. Among them, biological processes possess the advantages of having high removal efficiencies and low operational costs. However, it is difficult for microorganisms to directly adapt to the secondary effluent of CWW as the main components of the organics are refractory. As a solution, adding co-metabolic substrates, such as glucose, can provide sufficient carbon and energy sources for microbial growth, thus enhancing the ability of microorganisms to metabolize the recalcitrant organic compounds

through co-metabolism as well [12]. Jensen indicated that microorganisms used a growth substrate to improve the conversion efficiency of a non-growth substrate when the two types of substrates coexisted [13]. Liu and colleagues demonstrated that low concentration polyether wastewater could be effectively treated by acclimated activated sludge when sodium acetate was added as the carbon source [14]. Wang and colleagues found that when glucose was added as the carbon source to acclimated activated sludge for 180 days, the cyclohexane carboxylic acid could be steadily degraded by over 90% [15]. Sun and colleagues showed that adding glucose as a co-metabolic substrate to domesticate activated sludge in sequencing batch reactors (SBRs) could achieve the effective degradation of 2-chlorophenol at a concentration of 40 mg/L [16]. Therefore, adding easily biodegradable substrates/co-substrates to domesticate activated sludge can allow for the degradation of recalcitrant compounds in wastewater. However, the above-mentioned studies mainly focused on the degradation of a specific recalcitrant organic pollutant in simulated wastewater with the addition of a carbon source. Limited information is known about the degradation of the organic compounds in real wastewater. Moreover, the effect of the addition of an easily biodegradable substrate/co-substrate on the bacterial community structure has rarely been reported.

In this study, glucose was added as a co-substrate in the advanced treatment of recalcitrant organic compounds in CWW. The experiments included microbial acclimation and the operation of two parallel bioreactors after sludge acclimation. The scientific questions we addressed are: (1) how effective is the performance of acclimated sludge in the removal of recalcitrant organic matter; (2) how the diversity of the microbial communities differs before and after domestication; (3) whether the addition of glucose enhances the performance of the advanced treatment of secondary biochemical effluent of coking wastewater (SBECW). We believe that this study provides details of how the addition of a co-substrate could enhance the advanced treatment of CWW, which benefits the development of efficient biological treatment methods.

## 2. Materials and Methods

### 2.1. Experimental Setup and Procedures

The activated sludge was domesticated in an SBR, treating the effluent of the hydrolysis acidification-anoxic–oxic (HA-A/O) process in a CWW treatment plant in northwestern China. The excess sludge from the sedimentation tank of this plant was inoculated into the SBR as the source sludge. To ensure that the microbial community was well adapted to the characteristics of the SBECW and developed an ability to degrade recalcitrant compounds, microbial domestication was performed. The domestication reactor had an effective volume of 3 L (V) and its single operation cycle took 72 h. A cycle included feeding (10 min), aerobic reaction (65 h), a sedimentation phase (6.5 h), and discharge (20 min). The sedimentation phase provided an anoxic environment beneficial for endogenous post-denitrification to strengthen nitrogen removal [17]. The whole system of aeration intensity, water inflow, and drainage of the reactor were controlled by a rotameter, a peristaltic pump, and a siphon device, respectively. The SBECW in the influent was added at a gradient of 10% of effective volume (3 L) every 12 days (4 operation cycles), with the proportion of SBECW being 10% in the beginning of sludge domestication. For example, if the proportion of SBECW in the influent was 10%, 10% SBECW and 90% deionized water were mixed and added as the influent using a peristaltic pump. Glucose was added in the reactor at a concentration of 300 mg/L in this study [18]. Ammonium chloride ($NH_4Cl$) and sodium phosphate dibasic dihydrate ($NaH_2PO_4 \cdot 2H_2O$) were also added into the reactor in order to maintain the C: N: P ratio of the influent (including the carbon, nitrogen, and phosphorus pollutants that were originally present in the influent and the added carbon, nitrogen, and phosphorus substrates) at 100:5:1. The mixed liquor suspended solid (MLSS) concentration was kept at 3000–3500 mg/L, and the dissolved oxygen (DO) value was kept at $3.0 \pm 0.5$ mg/L. After acclimation, two parallel SBRs were seeded with the acclimated activated sludge ($A_0$ and $A_1$) and set up to evaluate the effects of glucose addition on SBECW treatment. $A_0$ was the

control reactor which was glucose-free and $A_1$ had the glucose addition. The dosages of glucose, $NH_4Cl$, and $NaH_2PO_4\cdot 2H_2O$ in $A_1$ were the same as those of the acclimation stage. The original MLSS concentrations of the two SBRs were 6300 mg/L. Other operational parameters were the same as those of the acclimation stage.

*2.2. Water Characteristics Measurement*

When the proportion of SBECW in the influent reached 50% and 100%, wastewater characteristics were analyzed. Influent and effluent chemical oxygen demand (COD) and ammonia-nitrogen ($NH_4^+$-N) concentrations were measured by the standard methods (Chinese SEPA, 2002) [19]. Total organic carbon (TOC) was analyzed by a TOC-VCPH analyzer (Shimadzu, TOC-VCPH, Tochigi, Japan). Ultraviolet-visible (UV-Vis) absorption and Ultraviolet 254 ($UV_{254}$) were measured by a DR6000 spectrophotometer (HACH, DR6000, loveland, CO, USA). The concentration of DO was measured by a YSI Pro 20 dissolved oxygen meter (YSI, Pro20, Yellow Springs, OH, USA). The pH value and temperature (T) of the reactor were measured by a portable pH meter (iSpring, TDS-pH-EC, Shanghai, China). MLSS of the sludge were determined according to the standard methods (APHA, 2005) [20]. The analysis of pollutant compositions in the wastewater was performed by using a gas chromatograph mass spectrometer (GC-MS) (Shimadzu, GC-MS 2010SE, Tochigi, Japan). The fluorescence scanning was performed by a Hitachi F-7000 three-dimensional fluorescence spectrophotometer (Hitachi, F-7000, Tokyo, Japan).

The molecular weight distributions of organic matter in the influent and effluent of the reactor were analyzed by using the continuous filtration method. The membrane (PL series, Millipore, Burlington, MA, USA) was soaked and rinsed three times (with the smooth side facing down) using ultrapure water with the addition of a little ethanol. The water samples were fractionated using six types of cellulose with molecular weight cutoffs of 1 k, 3 k, 5 k, 10 k, 30 k, and 100 k Dalton (Da), respectively. The effective surface area of the membrane was 31.75 $cm^2$. Afterwards, the COD concentration of each filtrate was determined by the standard method.

*2.3. DNA Extraction and Data Analysis*

The source sludge was collected for microbial analysis. When the SBECW's proportion reached 100% and the reactor was in a stable condition, the acclimated sludge was also collected for microbial analysis. The pellets were preserved at $-20\ ^\circ C$ for DNA extraction after centrifugation. DNA extraction was conducted by using the Qiagen 12888-100 DNA Kit (Qiagen, Washington, DC, USA). The V3 and V4 regions of the 16SrRNA gene were amplified using the 338F and 806R primers from the sludge samples. The purified PCR products were sent to Beijing Cell and Genome Corporation (Beijing, China) for high-throughput sequencing with the Illumina HiSeq platform. The raw sequences obtained were merged using Fast Length Adjustment of SHort reads (FLASH) [21] Low quality tags and ambiguous nucleotide chimeras were trimmed by using Quantitative Insights Into Microbial Ecology (QIIME) [22]. The chimera checking was performed using UCHIME [20]. Then the reads were assigned to operational taxonomic units (OTUs) with a 97% similarity threshold by Uclust [23]. The obtained representative sequence from each OTU was aligned to the SILVA bacterial database for taxonomic information.

Alpha diversity of the community based on Ace, Chao1, Shannon, and Simpson indices and OTU number were determined by Mothur version 1.30 software. The Bray–Curtis similarity index (Formula (1)) was used to evaluate the similarity between individual communities, which considers the number of common taxa and the abundance of each taxon, avoiding the influence of rare taxa.

$$C_{AB} = \frac{2\sum_{i=1}^{P}\min(y_{iA}, y_{iB})}{\sum_{i=1}^{P}(y_{iA} + y_{iB})} \tag{1}$$

*A* and *B* represent the source sludge and SBR, respectively. $y_{iA}$ and $y_{iB}$ represent the abundance of taxa in the source sludge and the SBR, respectively. A larger Bray–Curtis

similarity index suggests more similar communities. A paired t-test was done to make comparisons between the COD treatment efficiencies of $A_0$ and $A_1$ by Mothur version R.4.0.5 software.

## 3. Results and Discussion

### 3.1. Influent and Effluent Characteristics during Acclimation

When the proportion of SBECW was 50% and 100% in the influent, the concentrations of carbon and nitrogen pollutants were measured to determine the performance of the microorganisms during acclimation. The concentrations of COD, ammonia-nitrogen, and other water quality parameters of the influent and effluent are shown in Table 1. At an influent SBECW proportion of 50%, the concentrations of COD decreased from 112 mg/L to 97 mg/L, resulting in a removal efficiency of 13.39%. When the influent proportion of SBECW reached 100%, the concentrations of COD decreased from 249 mg/L to 222 mg/L, the removal rate of which was 10.84%. These results indicated that the microorganisms gradually adapted to the environment in the process of domestication. Based on these results, adding a co-substrate can promote the removal of refractory organic matter in CWW. It was consistent with the previous studies showing that the concentrations of organic matter and COD were reduced after sludge acclimation [24–26].

**Table 1.** Characteristics of the influent and the effluent of the SBR during the acclimation process.

| Wastewater | COD (mg/L) | $NH_4^+$-N (mg/L) | TOC (mg/L) | UV$_{254}$ (Abs) |
|---|---|---|---|---|
| Influent$_{50\%}$ | 112 ± 3 | 1.3 ± 0.4 | 23.23 ± 2.58 | 0.986 |
| Effluent$_{50\%}$ | 97 ± 3 | 1.1 ± 0.2 | 21.82 ± 2.12 | 0.953 |
| Influent$_{100\%}$ | 249 ± 4 | 4.3 ± 0.4 | 55.88 ± 2.94 | 2.416 |
| Effluent$_{100\%}$ | 222 ± 3 | 3.2 ± 0.3 | 52.53 ± 2.33 | 2.296 |

Abbreviations: Influent$_{50\%}$ represents the 50% proportion of the secondary effluent of CWW. Effluent$_{50\%}$ represents the 50% proportion of the secondary effluent of CWW from the acclimation system. Influent$_{100\%}$ represents the secondary effluent of CWW. Effluent$_{100\%}$ represents the secondary effluent of CWW effluent from the acclimation system.

### 3.2. Changes of Concentration and Components of Organic Matter during Acclimation

#### 3.2.1. UV-Vis Spectra

Ultraviolet-visible (UV-Vis) spectroscopy was used for the quantitative analysis of specific organic matter in the wastewater. The UV-Vis absorption bands in the wavelength range of 190–900 nm (Figure 1) were detected in both of the influent and effluent. The distributions of the UV-Vis bands for the influent and effluent did not differ much when the proportion of SBECW in the influent was 50% (Figure 1a). The absorption peaks between 200 and 225 nm were present in the influent and effluent, suggesting that conjugated molecules containing benzene rings were present in the wastewater [27].

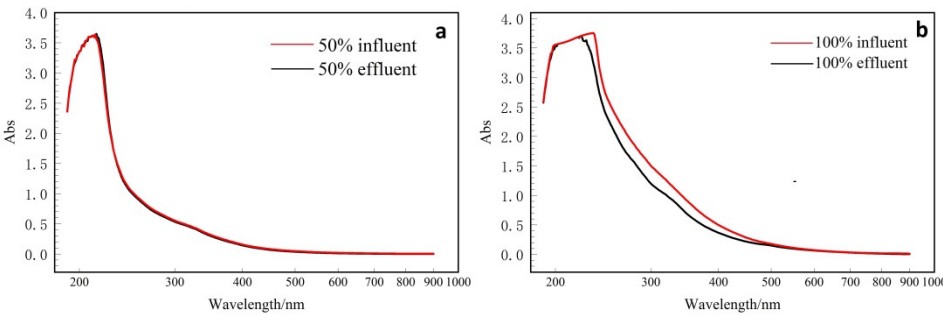

**Figure 1.** Results of the UV-Vis analysis for the influent and effluent in the acclimation system. (The proportion of SBECW in the influent and effluent was 50% (**a**); The proportion of SBECW in the influent and effluent was 100% (**b**)).

When the proportion of SBECW in the influent reached 100% (Figure 1b), the intensity of the UV-Vis spectra in the wavelength range of 220–230 nm displayed an obvious decrease after treatment, suggesting that the concentration of benzene series significantly decreased after treatment. In particular, the intensities of the absorption bands between 250 and 300 nm, and 300 and 370 nm gradually decreased in the effluent compared to those of the influent, respectively. This suggests the degradation of some polycyclic aromatic hydrocarbons (PAHs) and nitrogen-containing heterocyclic organic compounds in the reactor [27,28]. Together, these results indicate that the metabolic capacity of the microbial populations for organic pollutant degradation improved after acclimation.

3.2.2. Characterization of Fluorescing Organic Matter Fractions of CWW

The fluorescence images of wastewater were divided into five regions [29] (Table 2). In general, the peaks of shorter excitation wavelengths (<250 nm) and shorter emission wavelengths (<350 nm) are related to simple aromatic proteins such as tyrosine (Regions I and II). The peaks of shorter excitation wavelengths (<250 nm) and longer emission wavelengths (>350 nm) are related to fulvic-acid-like materials (Region III). The peaks of intermediate excitation wavelengths (250–280 nm) and shorter emission wavelengths (<380 nm) are related to soluble-microbial-byproduct-like material (Region IV). The peaks of longer excitation wavelengths (>280 nm) and longer emission wavelengths (>380 nm) are related to humic-acid-like organics (Region V) [30].

**Table 2.** Main fluorescence peaks and intensity of the Excitation–emission matrices (EEMs) of the wastewater.

| Fluorescence Region | Fluorescence Intensity (a.u.) (Peak Position) | | | |
| --- | --- | --- | --- | --- |
| | Influent$_{50\%}$ | Effluent$_{50\%}$ | Influent$_{100\%}$ | Effluent$_{100\%}$ |
| Region I | 230/350 (6463) | 230/355 (5979) | 230/345 (**** [a]) | 235/355 (4227) |
| Region II | 275/305 (2169) | 275/305 (2306) | 275/305 (8397) | 275/305 (1548) |
| Region III | 280/355 (4351) | 280/355 (4168) | 275/345 (**** [a]) | 280/355 (4918) |
| Region IV | 310/380 (3344) | 310/385 (3491) | 315/370 (6721) | 320/375 (3387) |
| Region V | 250/410 (2771) | 250/410 (2900) | 255/430 (5839) | 255/435 (3487) |

[a] (****) represents wastewater concentration that exceeded the maximum detection limit.

When the proportion of SBECW in the influent was 50%, the intensities of the fluorescence peaks of influent and effluent did not show obvious differences (Figure 2a,b), suggesting no obvious degradation of macromolecular organic matter in this acclimation stage. This phenomenon indicates that the ability of microorganisms to metabolize or co-metabolize organic matter was still limited.

When the proportion of SBECW in the influent was 100%, the intensities of the fluorescence peaks between the influent (Figure 2c) and the effluent (Figure 2d) did show obvious differences. Higher intensities of the fluorescence peaks were observed for the influent in Figure 2c than those of the effluent (Figure 2d). Peaks of the wastewater mainly occurred in regions I, II, and III. The organic compounds in the SBECW were mainly aromatic proteins and fulvic-acid-like materials. In regions I and III, the peak intensities decreased in the effluent compared to those of the influent, indicating that aromatic proteins and fulvic-acid-like materials were degraded after domestication. In addition, a blue shift of Regions II from the influent to the effluent was observed. This demonstrates that the number of aromatic rings and conjugated bonds with chain structure was reduced after activated sludge treatment, which results from the transformation of some aromatic compounds into smaller molecular compounds [31,32]. A possible reason for this is that the microbial populations could use glucose to metabolize or co-metabolize refractory organic matter [33] after acclimation.

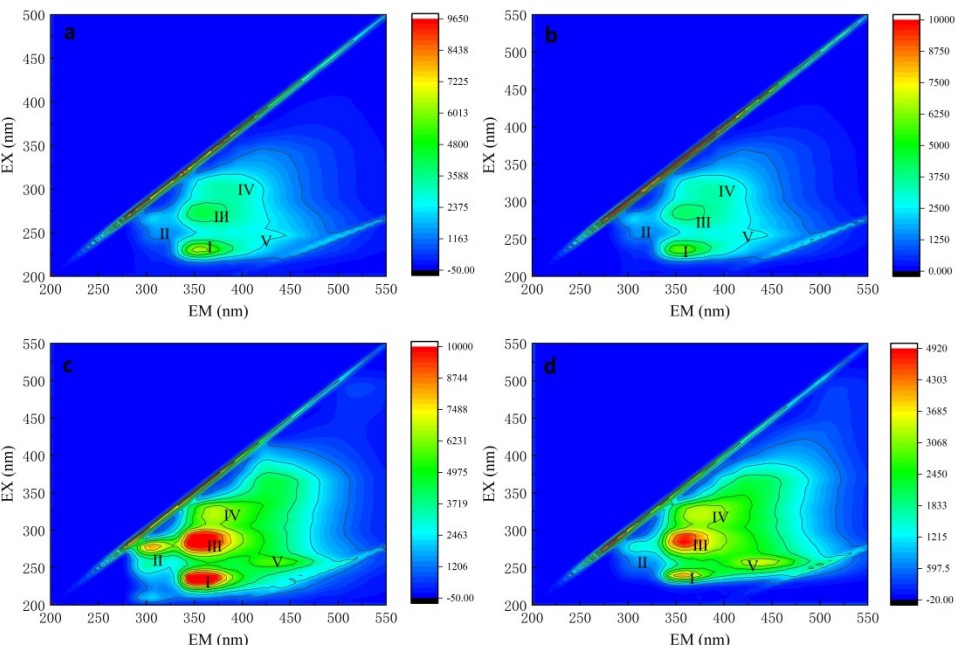

**Figure 2.** Three-dimensional fluorescence spectrums of influent and effluent in the acclimation system. (The proportion of SBECW in the influent was 50% (**a**); The proportion of SBECW in the effluent was 50% (**b**); The proportion of SBECW in the influent was 100% (**c**); The proportion of SBECW in the effluent was 100% (**d**); Emission wavelength (Em) from 200 nm to 550 nm (excitation slit: 5 nm band-pass) and excitation wavelength (Ex) from 200 nm to 450 nm (emission slit: 5 nm band-pass)).

### 3.2.3. Organic Components Revealed by GC-MS

GC-MS results are displayed in Table 3. Several soluble organics were present in the influent, including long-chain alkanes, benzene series, aromatics, halides, polycyclic aromatic hydrocarbons, and heterocyclics. After treatment, several heterocyclics and polycyclic aromatic hydrocarbon compounds were degraded, such as phenols, furans, indoles, naphthalenes, pyrazines, and carbazoles (Table 3). Nevertheless, compounds such as benzene series, halides, ethers, and amines remained in the effluent. In particular, the number of long-chain hydrocarbons increased from 9 to 16 after biological treatment. A possible reason for this is that ring-opening of the phenols and polycyclic aromatic compounds increased the number of long-chain hydrocarbons [34]. The performance of ketone removal was poor, possibly due to the biodegradation of complex molecules that may generate ketones as an intermediate that were not further degraded [35]. However, the number of toxic pollutants such as phenols, quinolones, and indoles in the effluent decreased [36,37], which indicated that some toxic compounds could be degraded by microorganisms in the domestication system.

### 3.2.4. Molecular Weight Distribution of Organic Matter

The molecular weight distributions of organic matter in the wastewater were analyzed by using the continuous filtration method. As shown in Figure 3, the molecular weight distribution of influent and effluent organic matter in the acclimation system exhibits a U-shape. Jin and colleagues also showed that the molecular weight distribution of influent and effluent organic matter in various biological reaction pools exhibits a U-shape [38]. Most organics in the influent and effluent showed molecular weights above 30 k Da or below 1 k Da (Figure 3). The proportions of organic matter with molecular weights over 30 k Da in the influent and effluent were 20.58% and 15.54%, respectively. The organic matter with molecular weights below 1 k Da in the influent and effluent accounted for 71.59% and 72.78%, respectively. Compared to the influent, the proportions of soluble, macromolecular organic matter decreased and the proportions of micromolecular organics

increased in the effluent. The organic matter of the influent in the molecular weight distribution ranges of 10–30 k Da, 5–10 k Da, 3–5 k Da and 1–3 k Da were lower than 5%, respectively. This reflects that these ranges of organic matter are difficult to degrade using a biological process [39]. However, the proportions of the organics matter in these weight distribution ranges in the effluent were all higher than those of in the influent (Figure 3). Therefore, it can be deduced that some macromolecular organic matter of a molecular weight above 30 k Da were converted or decomposed by microorganisms into substances of smaller molecular weights. Together, these results indicate that some soluble, macromolecular organic matter in the influent were converted to micromolecular organics during the acclimation process.

**Table 3.** GC-MS analysis results of the influent and effluent during acclimation.

| Type of Organic Matters | Number of the Investigated Organic Matters in Influent | Number of the Investigated Organic Matters in Effluent |
|---|---|---|
| Phenols | 5 | 2 |
| Quinolines | 2 | 1 |
| Furans | 6 | 2 |
| Indoles | 5 | 1 |
| Long chain alkanes | 9 | 16 |
| Benzenes | 8 | 7 |
| Amines | 3 | 2 |
| Halogens | 8 | 6 |
| Pyridines, Piperidines | 3 | 2 |
| Naphthalenes | 6 | 2 |
| Fluorenes | 2 | 1 |
| Indenes | 2 | 0 |
| Alcohols | 1 | 3 |
| Lipids | 4 | 1 |
| Ethers | 5 | 4 |
| Ketones | 1 | 1 |
| Pyrazines, carbazoles | 5 | 2 |

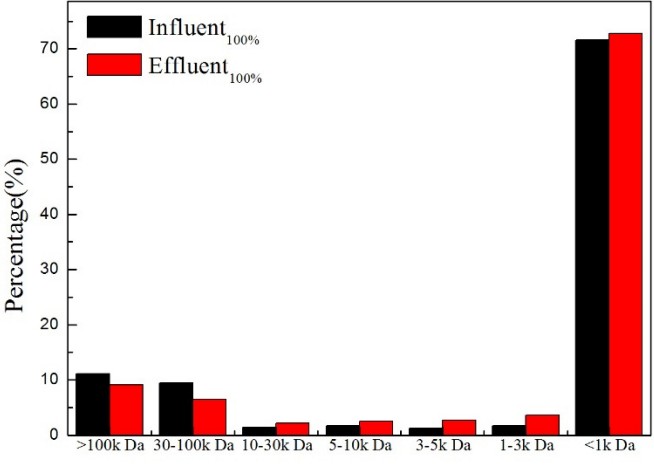

**Figure 3.** The molecular weight distributions of organic matter during acclimation.

### 3.3. Microbial Diversity of the Source and Acclimated Sludge

A total of 34,853 and 48,376 effective sequences were detected in the source sludge and the SBR sludge, which were clustered into 833 and 357 OTUs at a 97% similarity level, respectively. The coverage of each sample was 99.9%, indicating that the sequencing depth could represent the real diversity of the microorganisms in the samples. The alpha diversity indices were used to evaluate the richness and diversity of the microbial community in

each sample (Table 4). After acclimation, the ACE index decreased from 835 to 385 and the Chao1 index decreased from 840 to 397 for the communities. In addition, the Shannon index decreased from 5.00 to 3.40 and the Simpson index increased from 0.03 to 0.08 for the microbial communities. Together, these indicate that microbial diversity decreased after domestication. Similarly, a previous study reported that the Shannon index decreased from 6.58 to 5.30 and the Simpson index decreased from 0.05 to 0.02 after acclimation [18]. The decrease in microbial diversity after acclimation may be attributed the toxic and recalcitrant organic matter limiting the growth and metabolism of some microorganisms.

**Table 4.** Alpha diversity indices of the source community and acclimated community.

| The Sample Name | Number of Valid Sequence | No. of OTUs | ACE | Chao1 | Simpson | Shannon | Coverage [a] |
|:---:|:---:|:---:|:---:|:---:|:---:|:---:|:---:|
| In SBR | 48376 | 357 | 385 | 397 | 0.08 | 3.40 | 0.999 |
| Source sludge | 34853 | 833 | 835 | 840 | 0.03 | 5.00 | 0.999 |

[a] Coverage: Good's value, an estimation of the proportion of the population represented by the retrieved sequences.

The Bray–Curtis similarity index of the communities in the source sludge and the SBR was 0.29, which is lower than those of the investigated communities (about 0.60) reported by a previous study [40]. A reasons for this could be that the microbes in the source sludge, being without a strong degradation ability or high tolerance for recalcitrant compounds, gradually disappeared in the acclimated system.

*3.4. Microbial Composition of the Source and Acclimated Sludge*

Phylum distribution of the microbial communities in source sludge and SBR are displayed in Figure 4. At the phylum level, *Proteobacteria* (35.03%), *Cyanobacteria* (15.52%), *Firmicutes* (14.79%), *Bacteroidetes* (14.02%), and *Actinobacteria* (4.36%) were dominant in the source sludge. On the other hand, *Proteobacteria* (35.72%), *Planctomycetes* (18.94%), *Cyanobacteria* (17.19%), *Acidobacteria* (15.78%), and *Bacteroidetes* (6.53%) were dominant in the domesticated sludge. The relative abundance of *Planctomycetes* increased from 1.18% to 18.94% after domestication. A previous study reported that *Planctomycetes* participated in the degradation of polymeric organic compounds [41]. In addition, *Proteobacteria* were slightly more abundant in the domesticated sludge. This phylum was reported to be the most abundant phylum in several CWW treatment plants [42]. In addition, a previous study reported that *Proteobacteria* can participate in the degradation of phenol, polycyclic aromatic hydrocarbons, nitrogen-containing heterocyclics, and other recalcitrant organic compounds in CWW [43,44]. Together, these results indicate that the bacteria showing a capacity for the degradation of some toxic and refractory compounds were enriched after domestication.

The relative abundance of the dominant genera (with relative abundances over 1%) in the source sludge and in the SBR are displayed in Figure 5. Among them, the relative abundance of *Thauera* was 8.91% in the SBR, which was much higher than that of the source sludge (1.25%). *Thauera* plays an important role in the removal of pollutants including ammonium-nitrogen, organic matter, and aromatic compounds [45]. This genus was also shown to be a quinoline-degrading taxon in CWW [46]. Consistent with our study, a previous investigation found that *Thauera* was enriched (with a relative abundance of 6.26%) in the acclimated activated sludge in an SBR [47]. Moreover, the relative abundances of *Pesudomonas* and *Blastocatella* were 3.35% and 10.76%, respectively, in the SBR, which was higher than those found in the source sludge (0.06% and 0.02%, respectively). *Blastocatella* is considered to be a subfamily of *Acidobacteria* and is an aerobic heterotrophic bacterium, which can use oxygen as a receptor to transform the complex organic hydrocarbons and nitrogen-containing substances into small molecules [48]. *Pesudomonas* has the potential to degrade phenol and chlorophenol [49–51]. Overall, we found that the genera that possessed a high tolerance and strong metabolism for recalcitrant compounds were enriched after acclimation.

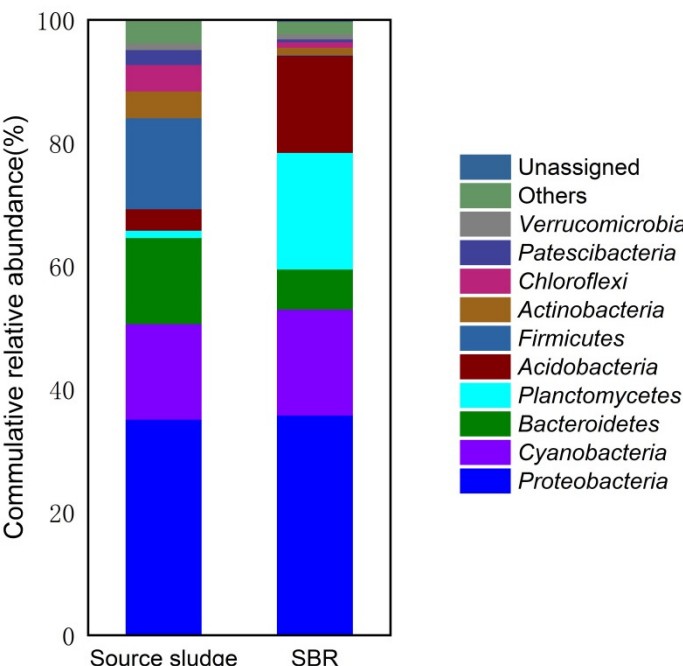

**Figure 4.** Cumulative relative abundance (%) of the detected phylum in the source sludge and SBR communities (a total of 10 dominant microbial phyla were detected in both samples).

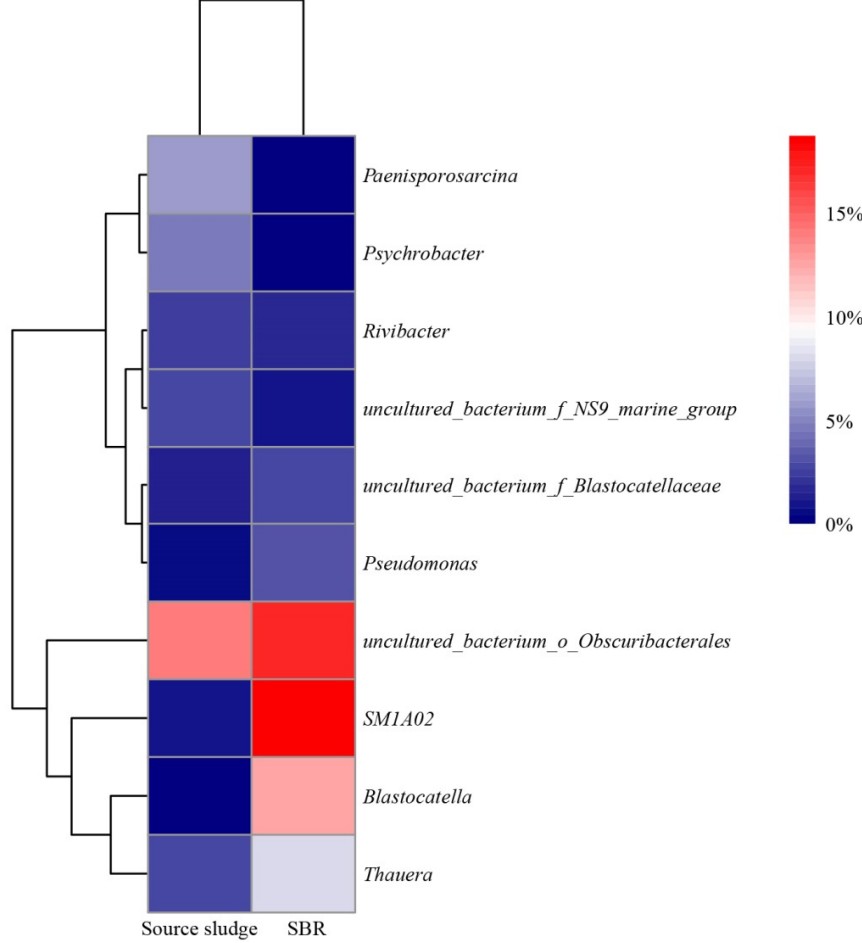

**Figure 5.** Relative abundances of abundant genera (a total of 10 dominant microbial genera were detected in the source sludge and SBR communities).

### 3.5. Effects of Glucose Addition on CWW Treatment

To decipher the effect of glucose addition on SBECW treatment by the acclimated activated sludge, the COD removal performance of two parallel SBRs ($A_0$ (glucose-free) and $A_1$ (glucose added)) were analyzed. The influent COD values and the removal efficiencies of the two bioreactors are shown in Figure 6. After 18 days in operation, the treatment efficiencies of COD in $A_1$ became stable and were between 15.78% and 16.78%, which was higher than those of the $A_0$ system (11.41–10.11%) ($p < 0.05$). A previous study also reported that without the assistance of co-metabolic substrates, the refractory and toxic organics presented in wastewater might inhibit microorganisms from degrading organic matter for a long time [14]. Therefore, it can be deduced that glucose functioned as a co-metabolic substrate and the degradation of some organic pollutants in the $A_1$ system was enhanced through microbial co-metabolism as compared to that in the $A_0$ system.

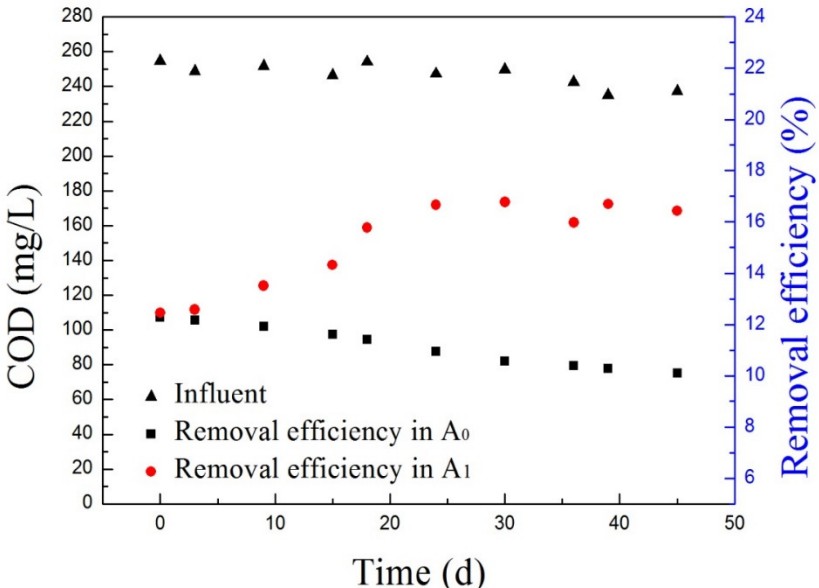

**Figure 6.** Variations of the COD removal performance and the influent COD values in parallel bioreactors. ($A_0$ (glucose-free) and $A_1$ (glucose added)).

### 3.6. Implications of Bioreactors with the Addition of a Co-Substrate in Wastewater Treatment

As stated in the Section 1, biological processes are advantageous due to their high removal efficiencies and low operational costs. In this study, a rough economic analysis was based on the cost of consumable materials of bioreactors with a co-substrate addition and a commonly applied advanced treatment technology—the catalytic ozonation process. The prices of the consumable materials were determined by consulting the reagent suppliers. According to the present study, the consumable materials of bioreactors with a co-substrate addition includes glucose, $NH_4Cl$, and $NaH_2PO_4 \cdot 2H_2O$. The prices of glucose, $NH_4Cl$, and $NaH_2PO_4 \cdot 2H_2O$ were \$156.42/t, \$262.78/t, and \$1095.27/t, respectively. Then, the cost of consumable materials for biological treatment is \$0.075/m$^3$ based on the addition doses used in this study. In the catalytic ozonation process, the efficiency of the removal of organic pollutants using catalytic ozonation depends on the catalyst. Therefore, the cost of the catalyst was analyzed in the reagent cost calculation. If activated alumina ($\gamma$-$Al_2O_3$) is selected as a catalyst, the catalyst addition is 70% of the effective volume and the bulk density of alumina is 0.75 g/mL [52]. When the wastewater treatment capacity is 50 m$^3$/h and the hydraulic retention time (HRT) is 90 min [53], the required catalyst is 39.357 t. In general, the service life of the catalyst is about three years. Assuming the price of the catalyst is \$3128.36/t, then the consumable materials cost of catalytic ozonation is \$0.094/m$^3$, which is higher than that of the biological process. Moreover, the operation cost of the catalytic ozonation process also includes the consumption of electricity and

liquid oxygen. The electricity cost of an ozone generator is relatively high compared to that of the aeration needed in the biological treatment. Furthermore, the catalytic ozonation process may show the disadvantage of catalyst instability after its long-term operation [53]. When the biological method with a co-substrate addition is applied to advanced CWW treatment in full-scale plants, methanol, which is a common product in a coking plant, can be used as the external carbon source instead, which reduces the material cost. Due to the strict transportation and storage conditions of methanol, glucose was selected as the carbon source in this study. Moreover, the reagent and operation cost of the biological method with a co-substrate addition can also be further reduced by the modification of the biological treatment processes. Therefore, the biological method with a co-substrate addition is a promising technology for the advanced treatment of industrial wastewater.

## 4. Conclusions

In this study, glucose was added in an acclimated system at a concentration of 300 mg/L to enhance the biological treatment of the secondary effluent of coking wastewater. The removal of organic pollutants was achieved after microbial acclimation, with decreases in the concentrations of COD, TOC, and $UV_{254}$ detected and the recalcitrant organic matter, such as polycyclic aromatic hydrocarbons and nitrogen-containing heterocyclics, being removed. The genera with greater tolerance to and degradation capacity for recalcitrant compounds, including *Blastocatella* (10.76%), *Thauera* (8.91%), and *Pesudomonas* (3.35%), were enriched in the acclimated system. To further investigate the effect of glucose addition on SBECW treatment by the acclimated activated sludge, the COD removal performance of two parallel bioreactors was analyzed. Results indicated that the addition of a co-metabolic substrate effectively improved the COD removal efficiency in $A_1$ as compared to $A_0$. The biological process with a co-substrate addition is more economical than the catalytic ozonation process for the advanced treatment of CWW. Moreover, the reagent and operation cost of the biological method with a co-substrate addition can be further reduced by modifying the bioreactors. Collectively, the biological method with a co-substrate addition enhanced the advanced treatment of CWW, and is a promising technology for the advanced treatment of industrial wastewater.

**Author Contributions:** N.L. did data analysis and wrote the manuscript draft. Y.X. was one of the experiment designers and revised the manuscript. X.H. supervised the data analysis. W.L. did data pretreatment. L.Y. performed data measurement. X.W. gave some suggestions for data analysis. Y.Q. gave some suggestions for manuscript revision. R.Y. helped design the experiments. X.G. helped in reviewing the manuscript. All authors have read and agreed to the published version of the manuscript.

**Funding:** The Fundamental Research funds of Central Universities of China, University of Mining and Technology (Beijing) and the National Key Research and Development Program of China (2017YFC1503101).

**Institutional Review Board Statement:** Not applicable.

**Informed Consent Statement:** Not applicable.

**Data Availability Statement:** Data sharing not applicable.

**Conflicts of Interest:** The authors declare no conflict of interest.

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
