# Peer review of "Glucose Addition Enhanced the Advanced Treatment of Coking Wastewater"

_water, doi:10.3390/w13233365_

Round 1

Reviewer 1 Report

The authors have investigated the treatment of CCW in a similar method to other researchers. While the results of the study is adequately presented, I would, however, have liked to see an economic and operational comparison with established treatment processes and the one proposed in this study. At the end of the day, economics often drive which process is chosen. Therefore a section on the comparative study is needed.

Reviewer 2 Report

The manuscript describes how the phenolic compounds of coking wastewater degraded in laboratory experiments

First in title change CWW to coke wastewater or coking wastewater! CWW has many other meanings! Coke is much more common than coking.

It is difficult to follow the paper since the experiment design is not described well. Did you have two bioreactors (or only one)? If you had two reactors did the other contain only coking wastewater and the other coking wastewater and glucose as a co-metabolic compound? Did you add the 100 % wastewater (+ biphosphate and ammonium chloride) to the experiments of 100 %? What about 50 % wastewater; was the dilutant water if was it tap water, some surface water or distilled water, or something else (+ biphosphate and ammonium chloride)? How did you mix wastewater and the dilutant (all at the beginning or  what means the sentence starting on line 82?) If the number of parallels was two, did you made the experiment one after the other meaning that there was some difference in the quality of influent? Describe this part much better!

What is the novelty of this paper?

In the abstract use UV absorption!

You use the word refractory organics. In many textbooks, those compounds are described as recalcitrant compounds.

You have some errors. The microbial genera in abstract and text are Thauera and Pseudomonas. In Fig. 4 they are correct. Correct also space before [ since this is incorrect since the line 32. Also previous is incorrect at least in lines 234 and 237.

In line 95: how many parallels?

Fig 1 omit water in explanations!

Fig 2 explain EM and EX! This figure is difficult to read since the scalings between c and d are very different. Discuss this!

Fig 5: What you would like to inform with Fig 5? I have to say that I cannot understand this. The differences between influent and effluent are very small and P-values are small. It is impossible to know do the sludge and sequence batch reactor sludges present the samples of this paper or some general samples. It is impossible to understand how these results have been got and how for instance sludge or wastewater could increase or decrease cardiovascular diseases. I would omit totally this figure. It highly overestimates the results of this paper.    

Fig 6: Explain A0 and A1!

Table 1. What should be compared? If influents and effluents turn this as Table 3.

In Table 3 you present the numbers of some mainly cyclic metabolite groups in influent and effluents (all must be in plural). The number of compounds over detection limit concentration can describe tentatively wastewater but not well its toxicity since different compounds with similar molecular weights can have very different toxicity (think about different mono-chlorophenols or still better di-chlorophenols!)  The respiration rate or biological oxygen demand of influent and effluent could give some evidence of ecotoxicity, but you did not do this.

In the conclusions, was your glucose concentration optimal? Would it be better to add it once or daily since glucose may be consumed rather rapidly?

Your references are partly incorrect. See the instruction!

There should not be , after the last author.

Use italic for the scientific names in references!

 Ref. 13 should be Jensen, In some cases, you have first family name and sometimes the initials. See what should be capitalized and what should not be capitalized. 

Reviewer 3 Report

Mocne strony:
Ogólny przeglÄ…d badania.
Wykorzystano świeżą literaturę.

SÅ‚abe strony:
Chociaż tematem publikacji jest Zaawansowane leczenie CWW wzmocnione dodatkiem glukozy poprzez ko-metabolizm drobnoustrojów, prezentacja wyników oraz ich analiza i dyskusja sÄ… dość skromne i zostaÅ‚y zawarte w zaledwie 9 wierszach na koÅ„cu manuskryptu.
Brak replikacji badania.
Nie przeprowadzono analizy statystycznej wyników.
Dość płytka dyskusja.
Badanie przeprowadzono wyłącznie w skali mikrolaboratoryjnej.
Nie byÅ‚o dyskusji na temat wdrożenia metodyki do peÅ‚nowymiarowej oczyszczalni Å›cieków.

Pytania do autorów:
Na jakiej podstawie dobierano dodatki glukozy, azotu amonowego i fosforanu sodu oraz ich dawki? (wiersz 87)
Jak zmiana (zwiÄ™kszenie/zmniejszenie) dawki dodatków wpÅ‚ynie na efektywność oczyszczania Å›cieków?
Gdyby autorzy dodali 300 mg glukozy na 1 l Å›cieków, to na 1 m3 potrzebne byÅ‚oby 0,3 kg glukozy. Dlatego proszÄ™ mi powiedzieć, czy takie rozwiÄ…zanie można realnie wdrożyć w peÅ‚nowymiarowych oczyszczalniach Å›cieków SBR?
Drobne poprawki:
Wzór sumaryczny chlorku amonu (linia 86) jest zapisany obok azotu amonowego.
Przydatne byÅ‚oby uzupeÅ‚nienie informacji o technologii oczyszczania zastosowanej w oczyszczalni Å›cieków, z której pobrano oczyszczony osad (linie 71-73).
ProszÄ™ usuÅ„. Powtórzenie z rozdziaÅ‚u metodologicznego (linie 167-168, 192-193)

Round 2

Reviewer 2 Report

The paper has been improved:

But:  In the manuscript, there are seven figures but in the text, there are mentions only of six figures. The present Fig 6 on page 11/16 must be omitted. This figure cannot belong to this paper since these diseases etc. have not at all been studied in this paper and they are totally out of this topic. On the contrary, the Figure 7 is the figure referred to on lines 324 and 325 as Fig. 6. In the text, there is no mention of Fig. 7. Change the number to the figure!

The MDPI journals are nice to read and to review since the tables and figures are in the text in the place where they belong. The authors should put the tables and figures in their correct place so that the Table 1 is near the present line 169. See then that there is no new page inside figures or tables!

Line 89 %V is not usual. Try to find a clear way to say this! Do you mean that the volume increased by 10%.

Line 165 Don’t use so many decimals. 249 and 221 are more real.

In almost all places starting from 37 you must add space before [. In line 37 it should be heterocyclics [3, 4]. This error is in many places, correct and check that in proofreading.    

Reviewer 3 Report

Thank you for responding to all the comments. The work looks much better and has gained depth. I only have a few additional comments due to the changes:

Please provide the source from which you determined the prices for the materials in lines : 338-350.

Please reorganize the information in subsection 3.2.4. Molecular Weight of Organic Matters is expressed in ''k Dalton''. In this paper, you have used the abbreviation from the manuscript of Jin et al. 2017. please change it. And remove the double dot at the end of the subsection.
